A real-time PCR assay for quantification of parasite burden in murine models of leishmaniasis

Antonia Alejandro L. 1
http://orcid.org/0000-0001-9556-2361 Wang Liuyang 1
http://orcid.org/0000-0002-0113-5981 Ko Dennis C. 1 2 dennis.ko@duke.edu
1 Department of Molecular Genetics and Microbiology, Duke University , Durham, NC , USA
2 Division of Infectious Diseases, Department of Medicine, Duke University , Durham, NC , USA
Lipoldova Marie
Electronic publication date: 2018 Nov 9
Publication date: 2018
Volume: 6
Electronic Location ID: e5905
Received 2018 May 9; Accepted 2018 Oct 9
Copyright: © 2018 Antonia et al.
Copyright year: 2018
Copyright holder: Antonia et al.
License: This is an open access article distributed under the terms of the Creative Commons Attribution License, which permits unrestricted use, distribution, reproduction and adaptation in any medium and for any purpose provided that it is properly attributed. For attribution, the original author(s), title, publication source (PeerJ) and either DOI or URL of the article must be cited.
License URL: https://creativecommons.org/licenses/by/4.0/

Keywords: Quantitative real-time PCR, Leishmania, Leishmaniasis, DRBD3, Mouse, Parasite burden

Funding: Dennis Ko from Duke University This work was supported by start-up funds for Dennis Ko from Duke University. The funders had no role in study design, data collection and analysis, decision to publish, or preparation of the manuscript.

==============================
Eukaryotic parasites in the genus Leishmania place approximately 350 million people per year at risk of disease. In addition to their global health significance, Leishmania spp. have served as an important model for delineating basic concepts in immunology such as T-helper cell polarization. There have been many qPCR-based assays reported for measuring parasite burden in humans and animals. However, these are largely optimized for use in clinical diagnosis and not specifically for animal models. This has led several of these assays to have suboptimal characteristics for use in animal models. For example, multi-copy number genes have been frequently used to increase sensitivity but are subject to greater plasticity within the genome and thus may confound effects of experimental manipulations in animal models. In this study, we developed a sybr-green based quantitative touchdown PCR assay for a highly conserved and single-copy putative RNA-binding protein, DRBD3. With primers that share greater than 90% sequence identity across all sequenced Leishmania spp., we demonstrate that this assay has a lower limit of detection of 100 fg of parasite DNA for Leishmania major, L. donovani, L. venezuelensis, and L. panamensis. Using C57BL6/J mice, we used this assay to monitor parasite burden over 1 month of infection with two strains of L. major (Seidman and Friedlin), and L. venezeuelensis. These characteristics rival the sensitivity of previously reported qPCR based methods of parasite quantitation while amplifying a stable, single copy gene. Use of this protocol in the future will lead to improved accuracy in animal based models and help to tease apart differences in biology of host-parasite interactions.

Introduction

Parasites in the genus Leishmania cause a spectrum of disease ranging from cutaneous leishmaniasis (CL) to visceralizing disease and are frequently used in many experimental animal models. Up to 1.6 million people each year are infected with one of the forms of leishmaniasis (Alvar et al., 2012). This significant impact of disease resulted in an estimated 980,000 disability adjusted life years (DALYs) in 2016 (GBD 2016 DALYs and HALE Collaborators, 2017). Further, these estimates are likely to significantly underestimate the burden caused by social and psychological stigmatization resulting from long term scarring (Bailey et al., 2017).

The outcome of Leishmania infection has been understood to depend on proper T-helper (Th) cell polarization since the late 1980’s when it was shown that a Th1 response promotes a healing immune response whereas a Th2 response leads to progressive, non-healing disease (Scott et al., 1988; Heinzel et al., 1989). Since then, animal models of leishmaniasis have continued to be used to characterize and understand many important aspects of the adaptive immune system. Despite these many advances in understanding immunologic concepts broadly, our understanding of the immune response to Leishmania spp. specifically is evolving to reflect greater nuances and complexities (Scott & Novais, 2016). Further, current treatment options for leishmaniasis remain prolonged, expensive, have variable efficacy, and have significant side effects, presenting an urgent need for novel therapeutics (Ponte-Sucre et al., 2017). In order to investigate these effectively, it is paramount to have optimal laboratory techniques for assessing disease progression in animals accurately, reproducibly, and under a variety of experimental conditions.

Methods to monitor disease progression during Leishmania infection in CL animal models have traditionally relied on measuring footpad swelling and using a limiting dilution assay (LDA) to quantify parasite burden (Sacks & Melby, 2001). While LDA remains a reliable and sensitive technique for quantifying parasite burden, these assays are labor intensive, can be technically challenging, and take several weeks to obtain final results. Recent advances employing genetic manipulation of parasites to express live reporter molecules such as mCherry or luciferase allow advanced monitoring of the parasite in real time and over a long-time course; however, they require an additional layer of genetic manipulation on the parasite and often require expensive equipment for visualization (Roy et al., 2000; Michel et al., 2011; Calvo-Alvarez et al., 2012). More recently developed protocols utilizing amplification of nucleic acids have been optimized for use in clinical diagnosis. Additionally, novel modifications to PCR based techniques, such as PCR-ELISA, hold significant promise in improving sensitivity and high throughput capacity of parasite detection (Kobets et al., 2010). To meet the demands of a clinical diagnostic assay, such as high sensitivity and the ability to discriminate between Leishmania spp., highly variable and multi-copy genes are most commonly used.

However, characteristics for clinical assays for Leishmania detection are not necessarily ideal for use in experimental animal models. The ability to differentiate between different strains is not required when the infecting species is carefully controlled and delivered during experimental infection. Additionally, the use of multi-copy genes which are known to be present in regions of relative genomic plasticity, may change during the course of infection. This at best adds unpredictable variation to the assay and at worst confounds the observation by inducing a systematic change across only certain experimental conditions. For instance, a comparative genomic study suggested that the high degree of variability of gene copy number across Leishmania spp. provides a potential mechanism to adapt to environmental change by altering gene expression (Rogers et al., 2011). Subsequently experimental studies have confirmed that Leishmania spp. regulate gene expression by altering gene copy number (Iantorno et al., 2017), and that this process can lead to drug resistance (Laffitte et al., 2016). Finally, it is particularly important to select genes without an active role in disease, as any studies to further understand these or related pathways are subject to the risk of mutations or gene copy expansions rendering PCR at these sites unable to accurately compare across experimental conditions.

Recently, several efforts have been made to standardize lab protocols for detection and quantification of parasites from clinical isolates (Cruz et al., 2013; Leon et al., 2017). However, similar comparative studies that emphasize assay traits optimized for experimental models of infection are lacking. Here we report a real-time quantitative PCR (qPCR) assay based on amplification of a single-copy, housekeeping Leishmania RNA binding protein (DRBD3) that is optimal for animal model studies. With touchdown cycling parameters, we were able to achieve a sensitivity of 100 fg per reaction which rivals most described PCR protocols for Leishmania quantification. Use of this assay in the future will facilitate studies elucidating mechanisms of immunity to Leishmania spp. and in monitoring efficacy of novel pharmaceutical interventions.

Materials and Methods

Multisequence alignment and sequence logo

Traits of the DRBD3 gene were analyzed in TriTrypDB using the Leishmania major Fd reference sequence (LmjF.04.1170) (Aslett et al., 2010). The reference sequence used in NCBI Blast tool to identify homologous sequences was based on the L. major SD75.1 sequence in order to correspond with the parasite DNA used to validate the assay in this study. The 11 identified homologous genomic DNA sequences were downloaded and aligned using ClustalOmega (McWilliam et al., 2013). The WebLogo tool was used to generate a sequence logo based on this alignment (http://weblogo.berkeley.edu/) (Crooks et al., 2004).

Primer design

Primers were designed using Primer-BLAST software (Ye et al., 2012). The gene for DRBD3 in the L. major (MHOM/SN/74/Seidman) was input. Five pairs of primers with similar melt temperatures were initially tested. After amplification by conventional PCR, three of these primer pairs resulted in non-specific amplification as detected by ethidium bromide detection in a 1% agarose gel. Amplicon size was verified by separating the PCR product in a 1% agarose gel, visualizing the product by ethidium bromide staining, and comparing to 100 bp DNA staining ladder (N3231S; New England Biolabs, Ipswich, MA, USA). Based on this initial screen, the primers reported in Fig. 1B were then used to amplify wildtype L. major (MHOM/SN/74/Seidman) promastigote DNA as described below.

Figure 1 Primer design and optimization of DRBD3 based qPCR for parasite quantification.

(A) Multisequence alignment based on 11 homologous sequences to L. major Sd. DRBD3 found using NCBI Blast. (B) Primer sequences and cycling parameters used. (C) DRBD3 primers amplify a 140 bp product specifically. Product visualized with ethidium bromide staining of a 1% agarose gel run and compared to New England BioLabs 100 bp ladder. (D) DRBD3 primers amplify diverse Leishmania spp. A representative plot of Ct value vs log dilution of parasite burden is shown. The average primer efficiency (±standard error of the mean) is indicated in parentheses for the following species: L. major Seidman (n = 6), L. major Friedlin (n = 5), L. venezuelensis (n = 5), L. panamensis (n = 3), and L. donovani (n = 3).

qPCR protocol for parasite quantification

After DNA isolation from each mouse tissue, 100 ng of DNA was used per reaction on a plate with a standard curve of cultured promastigote DNA ranging from 1 × 107 fg–100 fg per reaction. PCR reactions were set up in a final volume of 10 μl by adding five μL of ITaq universal SYBR Green supermix (Cat# 172-5121; BioRad, Hercules, CA, USA), 0.5 μl of each primer (one μM concentration), and 100 ng genomic DNA per sample. Cycling parameters are described in Fig. 1B. All PCR reactions were performed in triplicate. Samples were classified as having no Leishmania-specific amplification if melt temperature analysis revealed any products with a peak outside of the expected 84.7 ± 0.5 °C in any of the triplicate reactions. Primer efficiency for each reaction was calculated using the formula (10(−1m))−1, where m is the slope from a plot of the Ct vs log(parasite DNA) with serial 10-fold dilutions ranging from 1 × 107 fg to 100 fg total DNA per reaction.

Total parasite DNA was calculated by first using the average Ct for each reaction to interpolate the fg of parasite DNA per 100 ng of total DNA relative to the standard curve included on each 96-well plate. DNA (fg per reaction) was then multiplied by the factor used to dilute each sample to 100 ng per PCR reaction in order to get total parasite DNA (fg) per tissue harvested.

Parasites and culture conditions

Leishmania major Seidman (Sd.) (MHOM/SN/74/Seidman), L. major Friedlin (Fd.) (MHOM/IL/80/FN), L. donovani (MHOM/6D/62/1S), L. venezuelensis (MHOM/VE/80/H-16), and L. panamensis (MHOM/PA/94/PSC-1) were obtained from BEI Resources. Parasites were maintained in M199 media supplemented with 10% heat inactivated FBS and 0.2% hemin. Cultures were maintained by inoculating a new 10 mL culture with 200 μl of previous culture every 5 days. Infections in mice were performed with parasites that had been passaged less than three times in vitro.

Mouse infections

Wildtype C57BL6 mice from the Jackson Laboratory were maintained in the Duke Laboratory Animal Resource Breeding Core. All studies were approved under Duke University IACUC protocol A200-15-07. L. major (MHOM/SN/74/Seidman) parasites were prepared by washing 5 day old culture of promastigotes with Hanks Buffered Salt Solution (HBSS), counting by hemocytometer, and resuspending at 2 × 106 parasties per 50 μL of HBSS. A 27G 1/2 mL syringe with permanently attached needle was used to inoculate the left hind footpad with 50 μL of promastigote suspension. Mice were monitored at least twice weekly to track lesion development.

DNA was obtained using the Qiagen DNeasy Blood and Tissue Kit (Cat # 69504). In brief, tissue from the infected footpads and draining popliteal lymph node of each mouse was harvested. The contralateral footpad and lymph node was taken from each mouse to monitor for contamination. Tissue was placed into a clean 1.5 mL microcentrifuge tube. A total of 180 μL of buffer ATL and 20 μL of proteinase K was added to each tube prior to homogenizing the tissue with a bead beater and incubating samples at 37 °C overnight before proceeding as indicated by manufacturer instructions. DNA quantity and quality was assessed using the Take3 application on a Synergy H1 BioTek plate reader prior to use in the qPCR assay described. Thirty-three out of 35 uninfected footpads and non-draining lymph nodes, included as negative controls, did not produce a Leishmania specific product as determined by melt temperature analysis (84.7 °C).

Results

qPCR assay design for the Leishmania RNA binding protein, DRBD3

To design a qPCR assay with optimal characteristics for quantification of Leishmania parasite burden in experimental models, we first sought to identify a single copy gene essential to the parasite, but less likely to be actively involved in evasion of host defense. For this we chose putative Leishmania RNA binding protein (DRBD3), present in a single-copy on chromosome 4. The gene is classified as the double RNA binding domain 3 protein based on homology with DRBD3 in Trypanosoma brucei. Studies in T. brucei have identified a consensus binding sequence in the 3’UTR of mRNA transcripts and suggest that DRBD3 plays a role in modulating stress response (Fernandez-Moya et al., 2012; Das et al., 2015); however, there have been no functional studies of this protein in Leishmania spp. to date. The L. major entry for DRBD3 (LmjF.04.1170) on TriTrypDB indicates that the gene is constitutively expressed between parasite life-cycle stages, is not under immune pressure, and has minimal sequence variation between species (Aslett et al., 2010). RNA-seq experiments in both L. major and L. amazonensis have demonstrated little to no change in expression of this gene between the promastigote and amastigote stage consistent with the characteristics of a constitutively expressed housekeeping gene (Akopyants et al., 2004; Leifso et al., 2007; Aoki et al., 2017). Additionally, there were no identified epitopes in the Immune Epitope Database corresponding to DRBD3 peptides, suggesting that it is not likely to be influenced by host immune pressure (Vita et al., 2010). To verify the homology of DRBD3 between Leishmania spp. we used the sequence from L. major Sd. as a reference, and used NCBI blast software to identify all related sequences. This blast search yielded 11 hits of protein coding genomic DNA within the genus Leishmania. Using ClustalOmega we then created a multisequence alignment, and in combination with NCBI Primer Blast tool designed primers over a highly-conserved region (Fig. 1A). Both primers have complete conservation at 18 out of 20 nucleotides. This design allows using a single set of primers to PCR amplify a stable, housekeeping gene that is likely to be applicable across a broad range of Leishmania spp.

The assay was then validated with DNA from cultured L. major Sd. promastigotes. With touchdown PCR cycling parameters, one distinct product is visible by ethidium bromide staining, which corresponds with a single product by melt curve analysis at 84.7 °C (Figs. 1B–1C). Based on a standard curve extending across six orders of magnitude from 1 × 107 fg–100 fg of parasite DNA per reaction, the PCR had an average efficiency of 100.6% (n = 6) (Fig. 1D). The assay can detect DNA above this, but is no longer within the linear range. Using this same assay, DNA from geographically and phenotypically diverse parasites (L. major Fd., L. venezuelensis, L. panamensis, and L. donovani) were then validated and found to have the same limit of quantification (100 fg) and comparable primer efficiency (Fig. 1D). Of note, the L. panamensis sequence of DRBD3 has a mismatch at two out of 20 base pairs for each primer; however, this did not significantly impair the limit of quantification or primer efficiency. Therefore, we successfully developed a novel touch-down based PCR assay for a single-copy, housekeeping gene in Leishmania spp. that is able to efficiently detect as low as 100 fg of parasite DNA per reaction.

Validation of DRBD3 qPCR in vivo

We then tested the assay’s ability to monitor parasite burden in the murine model of CL. Wildtype C57BL6 mice were inoculated subcutaneously with 2 × 106 cultured L. major Sd. promastigotes in the left hind footpad. At 1 day and 34 days post infection, DNA was isolated from the infected and uninfected footpads as well as the corresponding popliteal lymph nodes.

The assay was able to accurately monitor parasite burden over this course of infection. Representative amplification and melting temperature plots are shown in Figs. 2A and 2B. In uninfected samples, nonspecific products were easily differentiated by a melting temperature outside of the expected 84.7 ± 0.5 °C. At 1 day post infection, six of seven mice had low, yet detectable parasite burdens in the footpad, but only two out of seven infected mice had detectable parasites in the draining lymph node. This is consistent with an expected delay in time for migration of Leishmania parasites to the draining lymph nodes. In both the draining lymph nodes and infected footpads, a significant increase in parasite burden was observed between 1 day and 34 days post infection, where all samples had Leishmania specific amplification (Fig. 2C).

Figure 2 DRBD3 primers are able to assess parasite burden from infected mouse tissue.

(A and B) Representative amplification plots and melt curve analysis from the draining lymph nodes of mice at 34 days post infection. Parasites amplified from infected tissue had melting temperatures corresponding with the expected 84.7 °C. DNA from uninfected tissue resulted in non-specific amplification at later cycles with melt temps outside of 84.7 ± 0.5 °C. Samples that generated any product outside of this range were classified as having no Leishmania specific amplification. (C) Quantification of total parasites in the draining popliteal lymph nodes and infected footpads at 1 and 34 days post infection with L. major Seidman (Sd.). At 1 day post infection, five lymph node samples had no Leishmania specific amplification and are not plotted. All samples from 34 days post infection had detectable Leishmania specific amplification. (D) DRBD3 quantification provides insights to disease pathogenesis in L. major Freidlin (Fd.) and L. venezuelensis infections. Comparison of footpad thickness and parasite burden in the draining lymph nodes and infected footpads at 36 days post infection. Four lymph node samples and one footpad samples had no Leishmania specific amplification and are therefore not plotted. P-values calculated by parametric Students T-test.

Having established that the primers work in vivo to monitor L. major Sd. infection, we next validated the assay on the traditional reference strain of L. major Fd. and with a strain of L. venezuelensis. Measuring footpad thickness with calipers demonstrate that at 36 days post infection there are significantly larger lesions in the L. major Fd. infected mice. However, using the DRBD3 qPCR method to quantify parasite burden revealed that the L. venezuelensis infected lesions harbored more parasites despite the lower overall degree of inflammation and swelling at the site of the lesion (Fig. 2D). The discordance is consistent with previous reports of L. major Fd. successfully clearing parasites, in contrast to infection with new world Leishmania spp. in the Leishmania subgenus, such as L. venezuelensis, which results in smaller, chronic and non-healing lesions (Scott & Novais, 2016). Therefore, this assay successfully monitors Leishmania spp. parasite burden in mouse models of CL.

Comparison of DRBD3 qPCR assay to other targets for quantification of Leishmania spp. infection in animal models

We compared the characteristics of the DRBD3 assay to other published qPCR assays. A wide variety of assays for Leishmania detection by PCR with diverse targets and characteristics have been described (Table 1), although most have been optimized from the perspective of clinical diagnostics.

Table 1 Comparison of described targets for qPCR based quantification of parasite burden in humans and animals.

Gene target	Copy number	Sybr/TaqMan?	Primer efficiency	Limit of quantification	Amplified from animal sample	Citation(s)	
Arginine Permease (AAP3)	1	TaqMan	1.051	10 fg	Mice	Tellevik et al. (2014)	
18 s rRNA	63–166	TaqMan/Sybr	0.831–0.942	10 fg	Humans; Sandflies	Shulzhenko et al. (2003), Van Der Meide et al. (2008), Bossolasco et al. (2003), Prina et al. (2007), Deborggraeve et al. (2008), Bezerra-Vasconcelos et al. (2011)	
DNA Polymerase	1	TaqMan/Sybr	0.934–0.9941	100 fg 0-500 parasites per 500 hepatic cells	Mice	Bretagne et al. (2001), Prina et al. (2007)	
Glucose 6 Phosphate Dehydrogenase (G6PD)	1	TaqMan/Sybr	0.504–0.957	100 fg 20 gene copies per rxn	Humans; Mice	Castilho et al. (2008), Prina et al. (2007)	
Glucosephopsphate isomerase (GPI)	1	TaqMan	NR	5.6 pg	Humans	Wortmann et al. (2005)	
Heat Shock Protein 70 (HSP70)	5–7	SYTO9	0.9237–0.9723	50 fg	Humans; Mice; Sandflies	Zampieri et al. (2016)	
Internal Transcribed Spacer (ITS1)	20–200	TaqMan	NR	0.2 parasites per sample	Humans; Dogs	Toz et al. (2013), Schonian et al. (2003)	
Kinetoplast Minicircle DNA (kDNA)	500–1,000	TaqMan/Sybr	0.79–1.05	10 fg	Humans; Dogs; Hampsters; Mice; Sandflies	Nicolas et al. (2002), De Paiva Cavalcanti et al. (2009), Francino et al. (2006), Mary et al. (2004), Jara et al. (2013), Srivastava et al. (2013), Bezerra-Vasconcelos et al. (2011)	
RNA binding protein (DRBD3)	1	Sybr	0.971	100 fg	Mice	This report	
Superoxide Dismutase B1 (SODB1)	1	Sybr	0.91	50 parasites	Mice	Ghotloo et al. (2015)	
Note:

Bretagne et al. (2001), Nicolas et al. (2002), Bossolasco et al. (2003), Schonian et al. (2003), Schulz et al. (2003), Mary et al. (2004), Wortmann et al. (2005), Prina et al. (2007), Castilho et al. (2008), Deborggraeve et al. (2008), Van Der Meide et al. (2008), De Paiva Cavalcanti et al. (2009), Bezerra-Vasconcelos et al. (2011), Jara et al. (2013), Srivastava et al. (2013), Toz et al. (2013), Tellevik et al. (2014), Ghotloo et al. (2015), Zampieri et al. (2016).

The most commonly used assays target multicopy genes. This produces advantages in regards to lower limits of detection and the potential to discriminate between species but come at the cost of uncertainty in gene stability at these plastic regions of the genome. Eighteen s rRNA is present in up to 166 copies per parasite (Ivens et al., 2005; Torres-Machorro et al., 2010), the internal transcribed spacer (ITS-1), found in between the small and large rRNA subunits between 40 and 200 (Cupolillo et al., 1995; Bensoussan et al., 2006), minicircle kinetoplast DNA at up to 10,000 copies (Yurchenko et al., 1999), and HSP70 at five to seven (Folgueira et al., 2007). These assays report limits of detection as low as 10 fg of parasite DNA per reaction. However, multicopy regions are often under significant change. For example, kDNA is particularly unstable in terms of copy number with reports of significant variation between species, strains, lifecycle stages, and clinical isolates (Mary et al., 2004; Weirather et al., 2011; Jara et al., 2013).

Single copy genes tend to be in more stable regions of the genome, but report higher limits of detection compared to multicopy gene assays and variable efficiencies between studies. Glucosephosphate isomerase (GPI), glucose 6 phosphate dehydrogenase (G6PD), Superoxide Dismutase 1 (SODB1) (Ghotloo et al., 2015), Arginine Permease (AAP3) (Tellevik et al., 2014), and DNA polymerase alpha (Croan, Morrison & Ellis, 1997) are all single copy genes for which Leishmania quantification by PCR has been described. The assay for GPI reports a higher limit of detection of 5.6 pg compared to other assays with limits of detection around 10–100 fg per reaction (Wortmann et al., 2005). Reported PCR efficiencies for the G6PD assays range from 50.4% to 95.7%. SODB1 and AAP3 described qPCR assays have competitive limits of detection and efficiencies (Tellevik et al., 2014; Ghotloo et al., 2015); however, it has become increasingly apparent that both of these genes are important virulence factors. AAP3 is upregulated in response to arginine shortages in host macrophages, and SODB1 deficient L. chagasi parasites demonstrate impaired survival in host macrophages (Plewes, Barr & Gedamu, 2003; Castilho-Martins et al., 2011; Goldman-Pinkovich et al., 2016; Muxel et al., 2017). This raises concern about the reliability of the assays during experimental manipulation; particularly in light of Leishmania spp. regulating gene expression through copy number variation (Laffitte et al., 2016; Iantorno et al., 2017).

Amplifying the gene target DNA polymerase alpha has similar characteristics to the DRBD3 assay. It is another example of single copy gene, which can be amplified at high efficiency with a limit of detection of 100 fg (Bretagne et al., 2001; Prina et al., 2007). Therefore, for monitoring parasite burden accurately and precisely in animal models, the DRBD3 qPCR assay and the DNA polymerase alpha assay (Prina et al., 2007) fulfill the optimal characteristics.

Discussion

Animal models of leishmaniasis have proven to be valuable in understanding basic immunological concepts and will be critical in future drug development programs to control this neglected tropical disease. The ability to accurately monitor parasite survival and replication in these models is paramount to properly understanding the biology of infection and monitoring the effectiveness of novel therapeutic interventions. The development of qPCR assays for parasite quantification has been widespread for clinical applications; however, there is no standardized qPCR protocol that is optimized specifically for use in animal models. Here we developed a novel touchdown qPCR assay of the single copy housekeeping gene, DRBD3, which has a limit of detection that rivals existing protocols while not being subject to changes in sequence or copy number that multicopy genes or genes required for virulence would be subject to.

Accurate measurement of parasite burden by qPCR is important for distinguishing between host and parasite mediated pathology. Disease progression of Leishmania animal models is traditionally done by a combination of measuring footpad thickness or directly measuring parasites through LDAs. Measuring the inflammation that results from infection by swelling at the site of inoculation is a valuable way to monitor severity of disease, but it is unable to distinguish between damage caused by overgrowth of the parasite and damage caused by dysregulated host derived inflammation. Studies showing that parasite growth does not perfectly correlate with lesion size demonstrate the importance of distinguishing between parasite and host mediated pathology (Hill, North & Collins, 1983; Bretagne et al., 2001). This is particularly relevant in the context of leishmaniasis where it is well documented that distinct disease manifestations are caused by both extremes of this spectrum (Scott & Novais, 2016). LDAs are useful for enumerating parasite burden, but are costly in terms of time and resources. Of note, the qPCR assay described here does require access to a real-time PCR thermocycler. However, given access to this equipment the protocol is significantly faster than the approximately 1 week required for parasites to grow using the LDA. After the initial overnight incubation used to isolate parasite DNA, using a qPCR approach to amplify parasite DNA allows determination of parasite burden in as little as 2–4 h.

The single copy gene DRBD3 is highly conserved across Leishmania spp., and the primers used in this assay bind to a region of DNA over which 90% of the residues are completely conserved. However, it should be noted that the divergence of the two nucleotide positions occurs at the division between the L. leishmania and L. viannia subgenera. The primers reported here are 100% conserved within the leishmania subgenus. Within the viannia subgenus, the sequences are also 100% conserved, and are only altered at two sites between the two genera for each primer. We demonstrate that despite these two divergent sites, the described primers work with high efficiency for parasites in the viannia subgenus (L. panamensis, see Fig. 1D). Further, the primers described here with optimal characteristics for parasite quantification in animal models could be adapted for use in developing iterations of PCR-based technology, such as PCR-ELISA (Kobets et al., 2010), in order to further improve sensitivity and capacity of the assay. The DRBD3 qPCR assay for Leishmania quantification is therefore also likely to be a robust assay for labs in that validation of the assay one time will allow for use in a wide range of studies modeling parasites with distinct disease phenotypes and from diverse geographic and evolutionary backgrounds.

Conclusions

Careful consideration of protocols used to quantify parasite burden in experimental models of leishmaniasis is essential to fully understand host-parasite interactions and for assessing the efficacy of novel therapeutic interventions. The DRBD3 assay described here will allow for consistent detection of parasites in an unbiased manner with a low limit of detection, facilitating discoveries in basic science and improvements in leishmaniasis treatment.

Supplemental Information

Supplemental Information 1 Raw data used to create Figures 1 and 2.

Click here for additional data file.

Additional Information and Declarations

Competing Interests

Author Contributions

Animal Ethics

Data Availability

The authors declare that they have no competing interests.

Alejandro L. Antonia conceived and designed the experiments, performed the experiments, analyzed the data, prepared figures and/or tables, authored or reviewed drafts of the paper, approved the final draft.

Liuyang Wang analyzed the data, contributed reagents/materials/analysis tools, approved the final draft.

Dennis C. Ko conceived and designed the experiments, authored or reviewed drafts of the paper, approved the final draft.

The following information was supplied relating to ethical approvals (i.e., approving body and any reference numbers):

Mouse studies in this manuscript are approved under Duke University IACUC protocol A200-15-07.

The following information was supplied regarding data availability:

Raw data are available in a Supplementary File.

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
