# Peer review of "A real-time PCR assay for quantification of parasite burden in murine models of leishmaniasis"

_PeerJ, doi:10.7717/peerj.5905_

## Round 0.1 · original submission · Major Revisions

The authors should pay more attention to unspecific amplification (comment of Reviewer 1) and modify the experimental design.
Please address both reviewers' comments and concerns.

Reviewer 1 ·

Basic reporting

The aim of this study was develop a sybr-green based quantitative PCR assay for a highly conserved and single copy, putative RNA binding protein, DRBD3. The article is interesting and authors are presenting somehow very novels results. The manuscript is clearly written in professional, unambiguous language. The introduction is well contextualized with relevant citations. The abstract needs more detail and structure conforms to PeerJ standards. The main text is structured according with PeerJ standards. The figures are relevant and present high quality. The table 1 can be published as supplementary file.

Experimental design

The design of this study is original and is within scope of PeerJ. The most important issue of the study was measure parasite burden from 1 to 34 days pos infection in fragment of lesion of mice with cutaneous leishmaniasis. However, the study was done with only animal model (C57BL6) infected with L. major. In contrast to L. braziliensis, the L. major produces an important parasite burden in the lesion. Furthermore, there was important unspecific DNA amplification of uninfected tissue samples. These unspecific DNA amplifications can have happened by big amount of DNA (100ng) in the 10 µL of reaction.

Validity of the findings

no comment

Additional comments

The article is interesting and authors are presenting somehow very novels results. The manuscript is clearly written in professional, unambiguous language and well contextualized with relevant citations. The abstract needs more detail and structure conforms to PeerJ standards. The use of qPCR to measure the Leishmania DNA is appropriate, but presents some limitations. One of them, there was of Leishmania DNA after healing of the lesion. Furthermore, the qPCR is expensive and require sophisticated equipments. It requires also long incubation (overnight) time for DNA extraction. These limitations can be better discussed in the paper. In the phrase (lines 269-272) “Using a qPCR approach to amplify parasite DNA allows determination of parasite burden in as little as 2-4 hours instead of waiting approximately a week for parasites to grow in the LDA” the authors need consider the extraction time. In the line 280-282, the author wrote that the primers could be adapted for use in PCR-ELISA to improve sensitivity and capacity of the assay. However, the sensibility of the qPCR is high than PCR-ELISA.

·

Basic reporting

The English language results adequate.

Check the references in the line 423-426, there is a repetition of the same reference.

In figure 1 legend, please verify the150bp PCR product length; submitting the primer’s sequences in primer-BLAST results in a 140bp PCR product length.

Regarding the figure 1C, show the entire agarose gel and report the type of DNA marker used for this experiment (report it also in “Primer design” section, line 109 of materials and method).

Experimental design

The authors state that amplifying DRBD3 gene overcome the problem of multicopy genes variability during the infection. This should be demonstrated by amplifying the template using other primers, such as those reported in table 1.

In the title of the manuscript, it has been reported “..in murine models of leishmaniasis”; to state that this model is generally suitable for different Leishmania species, they should validate the amplification protocol infecting with other species.

Line 178: please check the melting temperature.

In line 138 the amount of L. inoculated is 2x10e7 parasites in 50 µl, whereas in line 188 is reported an inoculation of 2x10e6 L.; please verify.

Please verify the text formatting of the whole manuscript: check also the space of the unit of measurement (e.g. line 113, 114).

Validity of the findings

In this manuscript, the authors show a new real-time PCR assay based on the amplification of the single-copy gene DRBD3 gene, suitable for animal model studies.
The manuscript is well written, the techniques are well described and there are no objective errors. However, the conclusion are too speculative and not completely supported by presented data.

Additional comments

In this manuscript, the authors show a new real-time PCR assay based on the amplification of the single-copy gene DRBD3 gene, suitable for animal model studies.

Making the appropriate revision, this manuscript could contribute to improve the use of the animal models in experiments involving infection with Leishmania spp.

---

## Round 0.2 · accepted · Accept

Dear Authors,

your manuscript has been accepted.

·

Basic reporting

no comment

Experimental design

no comment

Validity of the findings

no comment

Additional comments

The authors responded to the queries and all additions are very illustrative for the reader.
The revisited manuscript turns out to be complete and the conclusion are now completely supported by presented data. Therefore, I endorse the publication.